# Cryptic Diversity in Colombian Edible Leaf-Cutting Ants (Hymenoptera: Formicidae)

**DOI:** 10.3390/insects9040191

**Published:** 2018-12-12

**Authors:** Pepijn W. Kooij, Bryn M. Dentinger, David A. Donoso, Jonathan Z. Shik, Ester Gaya

**Affiliations:** 1Comparative Fungal Biology, Department of Comparative Plant and Fungal Biology, Royal Botanic Gardens, Kew, Richmond TW9 3DS, UK; E.Gaya@kew.org; 2Natural History Museum of Utah, University of Utah, 301 Wakara Way, Salt Lake City, UT 84108, USA; bryn.dentinger@gmail.com; 3Department of Biology, University of Utah, 201 Presidents Circle, Salt Lake City, UT 84112, USA; 4Departamento de Biología, Escuela Politécnica Nacional, Av. Ladrón de Guevara E11-253, Quito 17-01-2759, Ecuador; david.donosov@gmail.com; 5Centre for Social Evolution, Department of Biology, University of Copenhagen, Universitetsparken 15, DK-2100 Copenhagen, Denmark; jonathan.shik@gmail.com

**Keywords:** *Atta*, Attini, Cytochrome Oxidase I, fungus-growing ants, *hormigas culonas*, nutrition, pest management, phylogeny, species complex, taxonomy

## Abstract

Leaf-cutting ants are often considered agricultural pests, but they can also benefit local people and serve important roles in ecosystems. Throughout their distribution, winged reproductive queens of leaf-cutting ants in the genus *Atta* Fabricius, 1804 are consumed as a protein-rich food source and sometimes used for medical purposes. Little is known, however, about the species identity of collected ants and the accuracy of identification when ants are sold, ambiguities that may impact the conservation status of *Atta* species as well as the nutritional value that they provide to consumers. Here, 21 samples of fried ants bought in San Gil, Colombia, were identified to species level using Cytochrome Oxidase I (COI) barcoding sequences. DNA was extracted from these fried samples using standard Chelex extraction methods, followed by phylogenetic analyses with an additional 52 new sequences from wild ant colonies collected in Panama and 251 publicly available sequences. Most analysed samples corresponded to *Atta laevigata* (Smith, 1858), even though one sample was identified as *Atta colombica* Guérin-Méneville, 1844 and another one formed a distinct branch on its own, more closely related to *Atta texana* (Buckley, 1860) and *Atta mexicana* (Smith, 1858). Analyses further confirm paraphyly within *Atta sexdens* (Linnaeus, 1758) and *A. laevigata* clades. Further research is needed to assess the nutritional value of the different species.

## 1. Introduction

Relatively few taxa are still sustainably harvested from the wild for human consumption [1]. Of these taxa, molecular analyses to delimit species identity are increasingly uncovering evidence of inaccurate labelling, for instance in fish [2,3] and mushrooms [4]. The implications of this are important due to conservation concerns [5,6], nutritional value and the presence of allergens [7], as well as more legalistic issues concerning truth in advertising [2]. Leaf-cutting ants are still sustainably harvested from the wild and consumed by people. The term leaf-cutting ant encompasses up to 51 species classified within two ant genera—*Atta* Fabricius, 1804 and *Acromyrmex* Mayr, 1865—both included in the tribe Attini Smith, 1858 (Hymenoptera: Formicidae [8]). Leaf-cutting ants have an obligate mutualistic relationship with the fungus *Leucoagaricus gongylophorus* (Möller) Singer (1986), originating approximately 18 MYA [9,10,11,12], and are now dominant herbivores in the Neotropics [13]. They are also notorious pests, with mature colonies consuming up to 500 kg (dry weight) of plant material annually and have widespread economic impacts in local communities every year [14,15]. Throughout the 20th century, the genus *Atta* in was the subject of several taxonomical revisions using morphology characters (e.g., see [16,17,18]), but it was only in the early 21st century that a classification including four new phylogenetically distinct sections was proposed for the first time based on molecular evidence [19]. In this phylogeny, *Atta laevigata* (Smith, 1858) and *Atta sexdens* (Linnaeus, 1758) were found to be paraphyletic, and it was suggested that wider sampling would be needed to resolve these relationships.

Leaf-cutting ants can also provide nutritional value and have positive economic impacts. In several Latin-American countries, winged female leaf-cutting alates (among the largest of all ants) are also collected for human consumption as they disperse from their colony of birth for mating flights [20,21]. Throughout their Latin-American distribution, leaf-cutting ants have different common names, including ‘chicatanas’ in Mexico, ‘sauvas’ or ‘tanajura’ in Brazil, ‘hormigas culonas’ in Colombia, or ‘hormigas arrieras’ in both Colombia and Panama. While commonly eaten by indigenous tribes in Latin America, studies have mainly focused on species consumed in Mexico (e.g., see [22,23]), with some anecdotal studies focusing on Ecuador [24] and Colombia [20]. Insects form an important part of the diet of indigenous tribes, providing for example ca. 25% of the daily protein intake for men and 32% for women in a tribe in the Northwest Amazon [25]. An estimation of the nutritional value of Colombian *Atta* species found they contain on average 42–52 g protein per 100 g dry weight (corresponding mainly to the massive thoracic flight muscles), a protein to body mass ratio similar to that of game meat [20]. Today, fried leaf-cutting ant queens can often be purchased at local markets throughout several regions in Colombia and are also exported abroad [26]. With regards to the economic benefits, one study estimated that indigenous people from Arriaga, in Chiapas (Mexico), could potentially collect ca. 39 tons of *Atta* spp. per year per family [27]. In Colombia, prices have increased from ca. $44 USD per kg in 1993 [27] to $68 USD per kg today (equivalent to 200,000 COP, or 1/3 of a local minimum salary; personal observations on 26 November 2016 by P Kooij). This means that one family could collect ants worth a potential $1.5 million USD a year in 1993 and approximately $2.5 million USD now.

Given the potential benefits provided by *Atta* ants listed above, a flag must be raised with regard to the detrimental effects that unlimited collecting of local populations might pose. Firstly, over-collecting of certain species could lead to a population collapse (e.g., see [6]), given that most (ca. 95%) of queen ants fail to successfully found colonies under regular ecological conditions, the removal of so many additional queens by humans may have major negative long-term impacts on leaf-cutting ant populations [28,29]. Secondly, although leaf-cutting ants in general are considered pests, these ants generally serve an important role in nutrient recycling by decomposing plant material, liberating nitrogen and phosphorus back into the environment [30]. Moreover, some species can even be restricted to particular habitats outside the range of agriculture areas [14], thus reducing their harm to farmers. In summary, over-collection of certain species could have multiple effects on the ecosystem and local economy.

In Colombia, local distributors sell putatively monospecific packets, typically consisting of ants from different collection trips, and labelled as containing *A. laevigata* (Figure 1A). However, we are unaware of any scientific study exploring the validity of this packaging. We sought to confirm the identity of fried leaf-cutting ants, by sequencing the Cytochrome Oxidase I mitochondrial gene (COI), which has previously been shown to be useful to reveal cryptic diversity [31,32], of 21 individual queens arbitrarily sampled from a packet bought at a store in the locality of San Gil (Department of Santander, Colombia, N6.55533 W73.13993, Figure 1B). We compared these samples with other *Atta* sequences in our collections, and with sequences directly downloaded from GenBank [19,33], using a phylogenetic framework. As an additional note about this protocol, we note that while these packaged ants had been previously fried in oil, we were nonetheless able to acquire reliable sequence data, which indicates the results provided here could provide a template for further studies seeking to confirm the veracity of entomophagous claims.

## 2. Materials and Methods

### 2.1. Taxon Sampling

Twenty-one individuals of fried *Atta* female alates (“hormigas culonas”) were sampled from a packet bought at a local store in San Gil and used for DNA extraction and sequencing of the mitochondrial COI gene. In addition, fifty individuals belonging to various *Atta* species were gathered from either field or laboratory colonies previously collected near Gamboa (Panama, N9.11643 W79.70000)—one *Atta cephalotes* (Linnaeus, 1758), 47 *Atta colombica* Guérin-Méneville, 1844, and two *A. sexdens*. In addition, 251 COI sequences of different *Atta* species (generated in [19,33]) were retrieved directly from GenBank, so that the dataset covered 13 out of the 15 currently known species of *Atta* (see Appendix A for voucher information and accession numbers), including a total of 322 individuals. Finally, three species of the genus *Acromyrmex* (collected in Gamboa, Panama), were also sequenced and used as an outgroup—*Acromyrmex echinatior* (Forel, 1899), *Acromyrmex octospinosus* (Reich, 1793), and *Acromyrmex insinuator* Schultz, Bekkevold & Boomsma, 1998.

### 2.2. DNA Extraction and PCR Amplification

Genomic DNA was extracted using standard protocols by boiling the thorax or several legs in 150 µL of 10% Chelex X-100 (Sigma, St. Louis, MO, USA, C7901) at 99 °C for 60 min following the method from Walsh et al. [34]. A region of ≈1100 bp of the Cytochrome Oxidase I gene was amplified in two fragments of ≈400 bp and ≈700 bp using the primers Jerry [35] and Ben [36] and LCO1490 and HCO2198 [37], respectively. The latter primers were redesigned to align accurately with published *Atta* and *Acromyrmex* genomes [11,38,39]: LCO1490-PK 3′-TTTCTACNAATCAYAAAGAYATYGG-5′ and HCO2198-PK 3′-TAAACTTCNGGRTGACCRAARAATCA-5′. PCR amplifications were prepared for a 20 µL final volume using the following protocols: (1) for Jerry and Ben (Moreau) primers, reactions contained 10 µL VWR Red Taq DNA polymerase Master Mix (VWR International, Haasrode, Belgium), 0.4 µL for each of the 10 µM primers, 0.4 µL of 25 mM MgCl_2_, 7.8 µL ddH_2_O, and 1 µL of template DNA, and the program used included 5 min denaturing at 95 °C, followed by 25 cycles of 30 s denaturing at 95 °C, 45 s annealing at 60–55 °C with a touchdown of −0.2 °C per cycle, and 1 min extension at 72 °C followed by 20 cycles of 30 s denaturing at 95 °C, 45 s annealing at 55 °C and 1 min extension at 72 °C, and a final 10 min extension at 72 °C; (2) for LCO1490-PK and HCO2198-PK primers, reactions contained 10 µL VWR Red Taq DNA polymerase Master Mix (VWR International, Haasrode, Belgium), 0.6 µL for each of the 10 µM primers, 0.4 µL of 25 mM MgCl_2_, 7.4 µL ddH_2_O, and 1 µL template DNA, and used a program of 2 min initial denaturing at 95 °C, followed by 40 cycles of 30 s denaturing at 95 °C, 45 s annealing at 55 °C, and 1 min extension at 72 °C, with a final 5 min extension at 72 °C. PCR products were purified by adding 0.4 volumes of a master mix containing 3.33% Exonuclease I (20 U/µL; Thermo Scientific, Waltham, MA, USA, EN0582) and 6.67% Alkaline Phosphatase (1 U/µL; Thermo Scientific, EF0652) and ddH_2_O, followed by 15 min incubation at 37 °C and 15 min incubation at 85 °C. Purified products were sequenced bidirectionally using the PCR primers. Sequencing reactions were prepared in a 5 µL final volume containing 1 µL BigDye sequencing buffer (Big Dye Terminator Cycle sequencing kit, ABI PRISM version 3.1; PerkinElmer, Applied Biosystems, Foster City, CA, USA), 1.3 µL ddH_2_O, 0.3 µL MgCl_2_ (25 mM), 0.2 µL primer (10 µM), 0.2 µL BigDye3.1 (ABI), and 2 µL purified PCR product and used a program of 2 min initial denaturing at 95 °C, followed by 60 cycles of 10 s denaturing at 95 °C, 10 s annealing at 50 °C, and 2 min extension at 60 °C. Sequenced products were precipitated using ethanol precipitation as described by the manufacturer [40] and resuspended in 30 µL ddH_2_O before they were loaded on an ABI PRISM 3730 automated DNA sequencer (PerkinElmer, Waltham, MA, USA, Applied Biosystems).

### 2.3. Sequence Alignment

Sequence fragments were subjected to BLAST queries for an initial verification of their identities. They were then assembled with Geneious v8.1.7 [41]. Using USEARCH [42], we reduced the initial dataset from 325 sequences to 190 unique sequences (including the three outgroup samples), by removing any sequence more than 99% identical. This dataset was then aligned using different approaches. First, sequences were subjected to an automated alignment with MAFFT L-INS-I v7.215 [43] with default settings. This alignment was used as a reference alignment for subsequent validation of alignments obtained with the program PASTA v1.6.4 [44]. We then performed 23 separate analyses with PASTA using the original dataset increasing the maximum sub problem size in steps of five from 5 to 100 followed by three increments of 25 to a maximum of 175. Each resulting alignment was compared to the initial MAFFT alignment using the software FastSP v1.6.0 [45] and resulting SP scores (value of alignment accuracy compared to the reference alignment) and PASTA scores (value of alignment accuracy without a reference alignment) were plotted against the maximum sub problem size in order to select the optimal alignment (Appendix A).

After evaluating the SP and PASTA scores and resulting trees, the PASTA alignment with a maximum sub problem size of 100 was considered the most optimal and used for subsequent analyses. This alignment was then manually corrected by comparing the alignment to the nucleotide and protein sequences of the same region from the well-studied *Apis mellifera* Linnaeus, 1758 ([46], accession numbers L06178.1 and NP_008083.1, respectively). This allowed us to correct the alignment at the level of amino acids and to remove any residual gaps produced in the initial alignment. The alignment was manually corrected using amino acid-based color for guiding. All newly generated DNA sequences have been deposited in GenBank under accession numbers MK241586-MK241659 (see Appendix A for GenBank accession numbers per sample), and the alignment is available in TreeBASE under accession number 23442 [47]. The number of informative positions was calculated using WINCLADA [48].

### 2.4. Phylogenetic Analyses

The manually corrected alignment, was then used for phylogenetic reconstruction using two different optimization methods and three programs: Maximum likelihood analyses implemented with PASTA and RAxML v8.2.9 [49], and Bayesian analyses with MrBayes v3.2.6 [50]. We used PASTA to rerun the alignment with the maximum sub problem size at 100, while setting the ‘--aligned’ option to fix pre-aligned sequences and increasing the number of iterations from 3 to 10. Branch support was estimated with SH-like local support values using default settings. The program RAxML was then used to perform a maximum likelihood analyses on a three-partition dataset (COI/1st, 2nd, 3rd). The most likely tree was calculated with 1000 replicates and a GTRGAMMAI model of molecular evolution. Bootstrap branch support was obtained from 1000 replicates of ML bootstrapping conducted with the same settings and program.

Before running MrBayes, the AIC in jModelTest 2.0 [51,52] was used to estimate the optimal molecular evolutionary model for each of the three codon positions. A Generalized Time-Reversible (GTR, [53]) model with an estimated proportion of invariable (I) sites was selected for the first partition. The same model with invariable sites and a gamma distribution approximated with four categories was selected for the second and third codon positions. Prior distributions included a (1, 1, 1, 1, 1, 1) Dirichlet for the substitution rate, a (1, 1, 1, 1) Dirichlet for the state frequencies, a uniform (0, 200) distribution for the gamma shape parameter, a uniform (0, 1) distribution for the proportion of invariable sites, a uniform distribution for topologies, and an exponential (10) distribution for branch lengths in all partitions. Finally, a Bayesian inferred tree was reconstructed with MrBayes v3.2.6. Two runs with 10 chains were analysed for 30 M generations, with a tree sample frequency of 1000. The log-likelihood scores were graphically explored by plotting them against generation time with Tracer v1.6.1 [54] and set stationarity when log-likelihood values reached a stable equilibrium value [55] and when average standard deviation of split frequencies across runs dropped below 0.01. This was also verified with the AWTY program [56]. A burn-in sample of 25% was discarded for each run. The remaining 45,000 trees were used to estimate branch lengths with the sumt command in MrBayes. Internodes with bootstrap proportions ≥ 70%, SH-like local support values ≥ 80%, and Bayesian posterior probabilities ≥ 0.95 were considered strongly supported (Figure 2). Internodes with a bootstrap value >70% and a posterior probability <0.95 were also interpreted as well supported [57,58]. Internodal support resulting from posterior probabilities (Bayesian analyses) and bootstrap analyses (ML) were depicted on the MrBayes phylogeny using the box scheme (Figure 2 and Appendix A) as implemented in the ‘‘Hypha’’ module [59] of Mesquite [60]. The grouping of *A. laevigata* was based on genetic data alone; morphological identification is not possible, because legs, wings and often the heads (all used for morphological species identification) are removed in the frying process. Single origin polytomies of *A. colombica* and *A. cephalotes*, visible in Appendix A, were collapsed in Figure 2 for better visual representation. All scripts used for the analyses performed in this study can be found at on Github [61].

## 3. Results

The final alignment included 18 species (incl. 3 outgroup species), 190 sequences and a total of 1098 positions of which 329 were informative, thus enabling us to confidently place the fried queen samples in the existing *Atta* taxonomy. The majority-rule consensus tree resulting from the Bayesian analysis is shown in Figure 2 with internode support obtained with MrBayes, PASTA and RAxML indicated on branches with actual support values as well as colour coded boxes (see Figure 2 for details). Tree reconstructions obtained with each method separately, including branch lengths, are shown in Appendix A. Appendix A depicts the same tree as Figure 2 but without collapsed branches.

Our COI phylogeny was mostly concordant with an earlier *Atta* phylogeny including multiple genes [19], and we were able to reconstruct all sections within the genus as monophyletic except for section Archeatta, which was reconstructed as paraphyletic here (Figure 2). Additionally, we confirmed that *A. sexdens* is also paraphyletic, and includes the monophyletic clade containing *Atta robusta* Borgmeier, 1939, and possibly several other species currently registered as subspecies, although more data are needed to confirm this result. The main difference compared to the previously published phylogeny is the polyphyly of *A. laevigata* reported here for the first time, with two main monophyletic groups: (1) *A. laevigata* group 1, recovered in the section Epiatta and nested within a clade including the species *Atta bisphaerica* Forel, 1908, *Atta capiguara* Gonçalves, 1944, *Atta opaciceps* Borgmeier, 1939, *Atta saltensis* Forel, 1913 and *Atta vollenweideri* Forel, 1893 and (2) *A. laevigata* group 2, placed amongst the paraphyletic taxa of the section Archeatta with an unresolved position. Two additional sequences were reconstructed elsewhere, potentially due to sequence errors or misidentifications: One as sister to the species *A. capiguara* (EU848069) and another one recovered within the *A. sexdens* clade (EU848090). These results were consistent among all three tree reconstructions.

In summary, our results suggest that *A. laevigata*, as currently defined, might include more than one species, presenting a case of cryptic diversity in this lineage. Regarding the ‘hormigas culonas’ sampled from the bag bought in Colombia, we confirmed that the majority belonged to *A. laevigata* group 1. Since *A. laevigata* group 1 clade includes most *A. laevigata* samples, it is likely that it represents this species in a strict sense. If this is the case, then our results confirm that the identification given by the shop owners as well as a newspaper-clipping present at the shop was correct (Figure 1). However, we found in the same bag, two samples that were recovered outside the *A. laevigata* group 1. One sample grouped within the *A. colombica* lineage, in the *Atta* s. str. section, and the second sample was recovered in an unsupported position, close to the *A. laevigata* group 2 clade.

## 4. Discussion

Our molecular phylogenetic analysis revealed that ants contained in one single bag and sold as one species may actually represent three different species, one of them potentially new to science. However, the results also suggest that the majority of the collected ants sold as food in Colombia are accurately identified, despite occasional misidentifications. Considering that ants are collected in very large amounts, it may be expected that the number of different species collected on a regular basis is actually higher than what we were able to detect here. Furthermore, by combining our data with already published sequences, we were able to reconstruct most relationships of the known phylogeny of the genus *Atta* [19] with only one locus.

Our results also corroborate those of Bacci et al. [19], that *A. sexdens* is paraphyletic, while also showing for the first time that *A. laevigata* is polyphyletic, with the potential for cryptic diversity. While *A. laevigata* group 1 was recovered in the section Epiatta, to which the species previously was assigned, we also identified the new *A. laevigata* group 2 in the section Archeatta. Both Gonçalves [16] and Borgmeier [17,18] had previously identified several subspecies of *A. laevigata*, which supports a hypothesis of morphologically cryptic diversity. Furthermore, Borgmeier [18] stated that Pergande [62] in an earlier publication, described the type specimen for *Atta mexicana* (Smith, 1858) as *A. laevigata*, which suggests the potential of a species in the section Archeatta with similar morphology as the true *A. laevigata*. Additional data will be needed to disentangle the current status of this species. Although we were able to reconstruct most of the known *Atta* topology and species clades using a single marker (COI), support in deep branches of the tree was low, and several conflicts arose in deep internodes, due mostly to lack of characters. Recently developed high-throughput sequencing tools, such as Restriction site Associated DNA (RAD) markers [63] and Ultra-conserved elements [64,65], have been shown to be useful for resolving problematic phylogenies. These methods, in combination with a more complete sampling, should make it possible not only to improve the resolution of phylogenetic relationships within this genus, but also to help better understanding the cases of possible cryptic speciation.

Regarding the three genetically distinct species found in a single bag, it is plausible that misidentifications could vary throughout the collecting season with different species available throughout the year due to different times of mating flights [66]. The consequences of collecting different species as a food source are yet to be explored, but we propose that it may influence both the nutritional benefits from eating the ants, for instance if queens contain different fat contents in relation to their differences in mating flights (e.g., see [67]), as well as to the local health of the ants’ populations. In contrast to the thousands of dollars lost each year from ant-destroyed crops [14,15], thousands or even millions of dollars are gained out of the selling of ants as food [21,27]. Furthermore, *Atta* spp. are regularly used in traditional medicine by indigenous tribes in the Americas. The heads, and in particular the mandibles, of the soldiers have traditionally been used as stitches to close wounds [22], and it is believed among the indigenous tribes that eating ants can treat asthma and sore throats [68]. Given the diversity of ant chemistries [20], it is thus possible that different species are preferred for different applications.

The survival rate of *Atta* queens before they reach the age of 5 years old—the age needed for an *Atta* colony to start reproducing—is approximately of 1% [69,70]. Such a low survival rate is due to predation and diseases. The collection of alates, males and future queens that just left the parental colony to form new colonies, puts an extra pressure on the survival rate of this groups of ants, affecting rare species the most. The general conception of leaf-cutting ants as pests indicates that not much research has been done on the implications of over-collecting them. Many species serve, however, an important role in nutrient cycling through ecosystems [14] and removing them could cause unknown damaging effects.

## 5. Conclusions

In summary, our results show that occasional misidentifications occur by ant collectors, which potentially affects the nutritional value of the food items. We suggest that nutritional values for the different species of *Atta* should be investigated to test for potential effects on nutritional quality. We also confirm the paraphyly of *A. sexdens* and *A. laevigata* and suggest further research is needed to fully understand *Atta* diversity. We suggest that the diversity of *Atta* needs further exploration through morphological and genetic analyses on existing and new collections.

## Figures and Tables

**Figure 1 insects-09-00191-f001:**
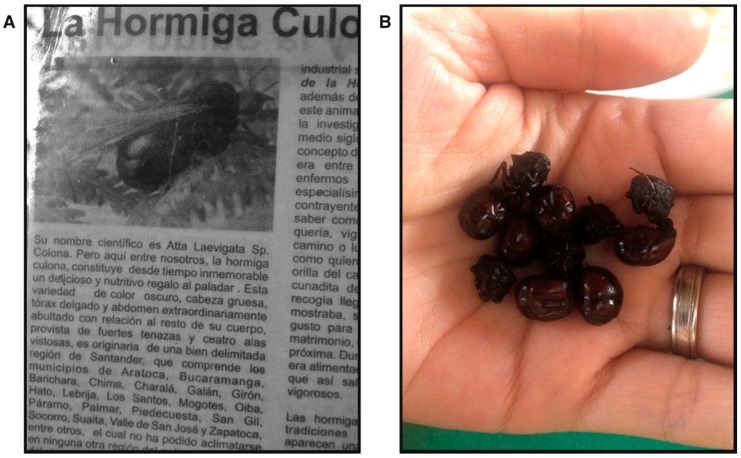
(**A**) Newspaper clipping found at the local store selling fried ants, i.e., “hormigas culonas”, as *A. laevigata*. The first two sentences translate to: “Its scientific name is *Atta laevigata*. But among us the “hormiga culona” is a delicious and nutritious gift for the palate, since time immemorial.” (**B**) Example of “hormigas culonas” sold at a local store in San Gil, Colombia.

**Figure 2 insects-09-00191-f002:**
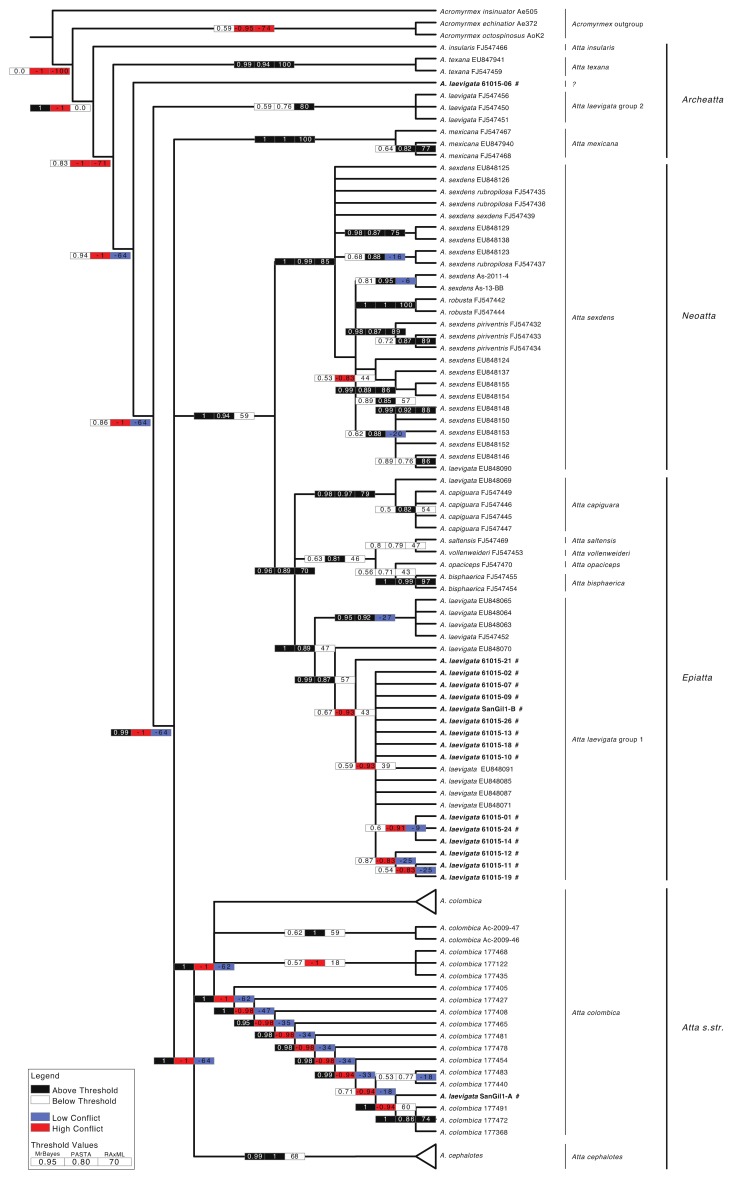
Phylogenetic relationships among 18 *Atta* members as reconstructed in the majority-rule consensus tree obtained with the Bayesian analyses. Three *Acromyrmex* species were used as the outgroup. The three-box grid associated with each internode indicates support obtained with the three sets of analyses: Bayesian analyses using MrBayes and maximum likelihood analyses obtained with PASTA and RAxML. Black boxes indicate relationships supported with posterior probabilities ≥ 0.95, SH-like local support values ≥ 80% or bootstrap support ≥ 70% in all three approaches; white boxes indicate resolved internodes in all analyses but with posterior probabilities < 0.95, SH-like local support values < 80% or bootstrap support < 70%; blue boxes indicate cases of unsupported conflict (i.e., posterior probabilities < 0.95, SH-like local support values < 80% or bootstrap support < 70%) when comparing the maximum likelihood trees to the Bayesian tree; and red boxes indicate significant conflict with posterior probabilities ≥ 0.95, SH-like local support values < 80% or bootstrap support ≥ 70% when comparing the maximum likelihood trees to the Bayesian tree. Cases of conflict occur when internodal support is not applicable because the branch was not recovered by the analyses (not present in the majority-rule consensus tree with all compatible groupings included; 1% threshold). Numerical support values are indicated inside the boxes. Taxon names highlighted in bold and marked with a ‘#’ indicate ‘hormigas culonas’ sampled from the Colombian bag. Triangle at branch tips represent collapsed clades of the same species. The full tree can be found in Appendix A.

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
