# Peer review of "Cryptic Diversity in Colombian Edible Leaf-Cutting Ants (Hymenoptera: Formicidae)"

_insects, 2018, doi:10.3390/insects9040191_

Round 1
Reviewer 1 Report
Line 2-3: It is a very widespread problem that the information on taxonomy and nomenclature is incomplete and consequently give little basis for validating taxonomic identifications. I thus propose to look in a very important paper by Packer et al. 2018 (Validating taxonomic identifications in entomological Research. Insect Conservation and Diversity 11: 1-12). Therefor I propose to include Hymenoptera, Formicidae in the title.
Line 14-27: Information in abstract should be self sufficient. Auctor and description year should be added to the scientific names.
General comment: Different journals have different styles. However, it is most common to use ( ) instead of [ ] in combination with auctor and year linked to scientific names. Since the paper treat taxonomic subjects, the reference to the original descriptions of the taxa mentioned should be added.
Line 43: [ Smith, 1858] comma added
Line 56: I propose to use species instead of ants in the headline since it is already stated that the paper is about ants.
Line 71-76: Text need to be changed. As far as I could see in the paper reffered to [32] the produced biomass available to each family in the area is 39 tons (page 256, table V). Each family did not collect 39 tons of ants, which would be fantastic. The value of 453 grams is said to be 20USD per 0.5 kg in the same paper. If "nowadays" means a particular year it should be mentioned. If the source is different it must be refered to. It is hardly possible that one family can collect ants worth 2.5 million USD per year. This must be the value of all ants collected? The entire paragraph should be rewritten!
Line 99-102: The scientific names must be in italic. Figure texts should be selfsufficient and auctor + year should be added. For information the name of paper and edition should be added.
Line105-109: The sentence is too long and Language should be improved.
Line114: "and" in the reference should be deleted and & used instead.
Line 115: Comma added [ Reich, 1793] and & should be used insted of and
Line 225-242: It would be nice if you included more information on what you call Atta laevigata group 1 and 2. Are these Groups Your creation based on Your DNA studies or do the have a taxonomic/morpohologic history?
Line 243 is not included in the manuscript
Line 244-245: Atta and Acromyrmex should be in italics.
268-270: The sentence is not well written.
General comments: I found the paper to be well written and interesting. The Language should be polished to shine more. The structure is fairly good. The weakest points is the precision in how nomenclature is used. The Methods and results of the DNA studies look healthy, although I have to admit I am not a specialist in this field. The coclusion is too short and should be extended. My conclusion is that the paper should be published with minor revisions.
Author Response
Comments and Suggestions for Authors
Line 2-3: It is a very widespread problem that the information on taxonomy and nomenclature is incomplete and consequently give little basis for validating taxonomic identifications. I thus propose to look in a very important paper by Packer et al. 2018 (Validating taxonomic identifications in entomological Research. Insect Conservation and Diversity 11: 1-12). Therefor I propose to include Hymenoptera, Formicidae in the title.
Agree, done
Line 14-27: Information in abstract should be self sufficient. Auctor and description year should be added to the scientific names.
Done
General comment: Different journals have different styles. However, it is most common to use ( ) instead of [ ] in combination with auctor and year linked to scientific names.
We have corrected nomenclature following the advice and regulations as stipulated in the International Code of Zoological Nomenclature (http://www.nhm.ac.uk/hosted-sites/iczn/code/) Chapter 11 Articles 51.2 and 51.3, which say “51.2. Form of citation of authorship. The name of an author follows the name of the taxon without any intervening mark of punctuation, except in changed combinations as provided in Article 51.3.” and “51.3. Use of parentheses around authors' names (and dates) in changed combinations. When a species-group name is combined with a generic name other than the original one, the name of the author of the species-group name, if cited, is to be enclosed in parentheses (the date, if cited, is to be enclosed within the same parentheses).” We, therefore, edited citations to have no parentheses for the scientific names on first appearance, unless the name comes from a recombination, for which parentheses are used. All citations now adhere to this Code as well as the http://antcat.org and http://www.speciesfungorum.org databases, which holds the most accurate citations for ants and fungi.
Since the paper treat taxonomic subjects, the reference to the original descriptions of the taxa mentioned should be added.
Where appropriate, full references were already given (see Gonçalves 1942, Borgmeier 1950 and 1959 and Pergande 1895). We, however, deemed unnecessary to do this for all species citations. Also, in the Code, mentioned above, there are no specific regulations regarding these issues, and it is not standard practice to do so.
Line 43: [ Smith, 1858] comma added
Done
Line 56: I propose to use species instead of ants in the headline since it is already stated that the paper is about ants.
On advice by another reviewer, the subheading has been removed
Line 71-76: Text need to be changed. As far as I could see in the paper reffered to [32] the produced biomass available to each family in the area is 39 tons (page 256, table V). Each family did not collect 39 tons of ants, which would be fantastic. The value of 453 grams is said to be 20USD per 0.5 kg in the same paper. If "nowadays" means a particular year it should be mentioned. If the source is different it must be refered to. It is hardly possible that one family can collect ants worth 2.5 million USD per year. This must be the value of all ants collected? The entire paragraph should be rewritten!
We thank the reviewer for pointing this out. The original table (V) in the referred paper was rather ambiguous regarding this information, and we agree with the reviewer that most likely was referring to the potential collection size, rather than the actual collection. Text has been rephrased accordingly to “With regards to the economical benefits, one study estimated that indigenous people from Arriaga, in Chiapas (Mexico), could potentially collect ca. 39 tons of Atta spp. per year per family [26]. In Colombia, prices have increased from ca. $44 USD per kg in 1993 [26] to $68 USD per kg today (equivalent to 200,000 COP, or 1/3 of a local minimum salary; personal observations on 26/11/2016 by P Kooij). This means that one family could collect ants worth a potential $1.5 million USD a year in 1993 and approximately $2.5 million USD now. ”
The increase in price was observed by Kooij on 26/11/2016 in Colombia, where 100g of ‘hormigas culonas’ was sold for 20,000 COP. We now added the date of the observation to the text and recalculated the price from 1993 to represent the price per kg.
Line 99-102: The scientific names must be in italic. Figure texts should be selfsufficient and auctor + year should be added. For information the name of paper and edition should be added.
Name is now in italic. However, we didn’t add author citation here as this is not suggested by the journal’s author instructions, the Code or the Packer et al 2018 paper mentioned by the reviewer above.
Line105-109: The sentence is too long and Language should be improved.
We thank the reviewer for the comment and split the sentence into two sentences as follows: “Twenty-one individuals of fried Atta female alates (‘hormigas culonas’) were sampled from a packet bought at a local store in San Gil and used for DNA extraction and sequencing of the mitochondrial COI gene. In addition, fifty individuals belonging to various Atta species were gathered from either field or laboratory colonies previously collected near Gamboa (Panama, N9.11643 W79.70000) – one Atta cephalotes (Linnaeus, 1758), 47 Atta colombica Guérin-Méneville, 1844, and two A. sexdens.”
Line114: "and" in the reference should be deleted and & used instead.
This citation has been changed to the more correct (Forel, 1899), as shown by the www.antcat.org database.
Line 115: Comma added [ Reich, 1793] and & should be used insted of and
Done
Line 225-242: It would be nice if you included more information on what you call Atta laevigatagroup 1 and 2. Are these Groups Your creation based on Your DNA studies or do the have a taxonomic/morpohologic history?
The grouping is based on the DNA study alone. In the frying process the legs, wings and often the heads are removed. Therefore, morphological identification is not possible emphasizing the need to identify the samples using genetic markers. We added a statement L278-280 “The grouping of A. laevigata was based on genetic data alone; morphological identification is not possible, because legs, wings and often the heads (all used for morphological species identification) are removed in the frying process.”
Line 243 is not included in the manuscript
Line 243 was the figure. Because of the use of track changes there are now several line number jumps.
Line 244-245: Atta and Acromyrmex should be in italics.
In the original word documents these are in italics, so I think something went wrong in the PDF generation.
268-270: The sentence is not well written.
This sentence has now been rewritten as “Furthermore, by combining our data with already published sequences, we were able to reconstruct most relationships of the known phylogeny of the genus Atta [11] with only one single locus.”
General comments: I found the paper to be well written and interesting. The Language should be polished to shine more. The structure is fairly good.
The native speaking co-authors have now gone through the manuscript with more detail to correct for this.
The weakest points is the precision in how nomenclature is used.
Given the reviewer’s suggestions, we implemented the nomenclature now correctly throughout the manuscript, as described in responses above.
The Methods and results of the DNA studies look healthy, although I have to admit I am not a specialist in this field. The coclusion is too short and should be extended. My conclusion is that the paper should be published with minor revisions.
The length of the conclusion was based on recent publications in the journal. We have added the following sentences: “We suggest that nutritional values for the different species of Atta should be investigated and the potential effects on food quality should be tested.” and “We suggest that the diversity of Atta needs further exploration through morphological and genetic analyses on existing and new collections.”
Reviewer 2 Report
Dear authors
I agree with value of your article. Your research is very novel.
I have minor points.
<Minor points>
#Abstract
L19-21: Here, we identified to species level 21 samples of fried ants bought in San Gil, Colombia, using COI barcoding sequences.
=> Here, species level 21 samples of fried ants bought in San Gil, Colombia, were identified using COI barcoding sequences.
L21-23: We extracted DNA of the 21 fried food samples using standard Chelex extraction methods, followed by phylogenetic analyses with an additional 52 new sequences from wild ant colonies collected in Panama and 251 sequences publicly available.
=> DNA of the 21 fried food samples were extracted using standard Chelex extraction methods, followed by phylogenetic analyses with an additional 52 new sequences from wild ant colonies collected in Panama and 251 sequences publicly available.
L26-27: We also suggest that the different species should be assessed separately for their nutritional value.
=> It was also suggested that the different species should be assessed separately for their nutritional value.
I consider that the word ‘I’ or ‘we’ are not suitable for the subject words in abstract.
#Keywords
Please add words ‘pest management’.
#Introduction
If you can, please introduce concrete example of agricultural pests and biotic resource.
#Discussion
If you can, please discuss pest management and using biotic resource. I hope that your article help development applied entomology.
#Whole
Your manuscript seems to need English correction by the native English speaker.
Author Response
Comments and Suggestions for Authors
Dear authors
I agree with value of your article. Your research is very novel.
I have minor points.
We greatly appreciate the constructive and encouraging remarks made about our manuscript by the reviewer. Having revised our manuscript in line with these comments, we believe it has been significantly improved.
<Minor points>
#Abstract
L19-21: Here, we identified to species level 21 samples of fried ants bought in San Gil, Colombia, using COI barcoding sequences.
=> Here, species level 21 samples of fried ants bought in San Gil, Colombia, were identified using COI barcoding sequences.
Following the reviewer’s advice, the sentence has now been changed to “Here, 21 samples of fried ants bought in San Gil, Colombia, were identified to species level using COI barcoding sequences.”
L21-23: We extracted DNA of the 21 fried food samples using standard Chelex extraction methods, followed by phylogenetic analyses with an additional 52 new sequences from wild ant colonies collected in Panama and 251 sequences publicly available.
=> DNA of the 21 fried food samples were extracted using standard Chelex extraction methods, followed by phylogenetic analyses with an additional 52 new sequences from wild ant colonies collected in Panama and 251 sequences publicly available.
Following the reviewer’s advice, the sentence has now been changed to “DNA was extracted from these fried samples using standard Chelex extraction methods, followed by phylogenetic analyses with an additional 52 new sequences from wild ant colonies collected in Panama and 251 publicly available sequences.”
L26-27: We also suggest that the different species should be assessed separately for their nutritional value.
=> It was also suggested that the different species should be assessed separately for their nutritional value.
Following the reviewer’s advice, the sentence has now been changed to “Further research is needed to assess the nutritional value of the different species.”
I consider that the word ‘I’ or ‘we’ are not suitable for the subject words in abstract.
In light of this we further changed L29-32 from “Our analyses showed that most samples are A. laevigata, however, one sample was identified as Atta colombica and another one formed a distinct branch on its own. Our analyses further confirm paraphyly within Atta sexdens and A. laevigata clades.” to “Most analysed samples corresponded to Atta laevigata (Smith, 1858), even though one sample was identified as Atta colombica Guérin-Méneville, 1844 and another one formed a distinct branch on its own, more closely related to Atta texana (Buckley, 1860) and Atta mexicana (Smith, 1858). Analyses further confirm paraphyly within Atta sexdens (Linnaeus, 1758) and A. laevigata clades.”
#Keywords
Please add words ‘pest management’.
Done.
#Introduction
If you can, please introduce concrete example of agricultural pests and biotic resource.
Although we appreciate the reviewer’s suggestion; the manuscript is not aiming at discussing agricultural pests but rather at the untapped potential diversity of the genus Atta and possibly other leaf-cutting ants. Nevertheless, we have highlighted their role as pests by adding a statement in L60-62 “They are also notorious pests, with mature colonies consuming up to 500 kg (dry weight) plant material annually and have widespread economic impacts in local communities every year [13,14].”
#Discussion
If you can, please discuss pest management and using biotic resource. I hope that your article help development applied entomology.
As we stated in the response above, we appreciate the reviewer’s suggestion and interest in developing ideas for pest management. However, the manuscript is not aiming at the agricultural pests side of leaf-cutting ants but rather at the potential diversity of the genus Atta and other leaf-cutting ants.
#Whole
Your manuscript seems to need English correction by the native English speaker.
The native speaking co-authors have now gone through the manuscript with more detail to correct for this.
Reviewer 3 Report
General comments
This manuscript made a survey on the diversity of leaf-cutting ant species in the market, which are produced as a human food. There is a serious concern on their data. The sample ants for DNA sequencing were fried. DNA could have a strong damage from the temperature 95C, while the oil temperature for frying is generally over 500C. They should provide the proof first that frying process did not harm DNA sequencing, or they should have used the samples from the market before they had been cooked. Except the sample, the methodology is fine, and all other parts were well written.
Here are numbers of suggestions, which might help to improve the manuscript.
Through manuscript, suggest to use the different English words for describing human society and ant ecology.
Ex. Authors use 'local population'(line16) to describe the human society, however it is better just to say 'local people'.
In abstract
Please focus the topic to leaf-cutting ant as a food source. Covering topic about them as a pest, a forest depredator and a food source is not necessary in abstract.
In introduction
Probably better without subtitles, or add subtitle to all paragraphs.
Line 32-44, suggest to replace it to the discussion.
Line 45-55, suggest to make the general explanation in 2-3 lines.
In materials and methods
Line 108, 114, 115, please correct the reference style.
In discussion
Probably better without subtitles, or add subtitle for all paragraphs.
In references
Suggest to reduce the references. There are too many references, which are only for general explanations of ecology of leaf-cutting ants.
Author Response
Comments and Suggestions for Authors
General comments
This manuscript made a survey on the diversity of leaf-cutting ant species in the market, which are produced as a human food. There is a serious concern on their data. The sample ants for DNA sequencing were fried. DNA could have a strong damage from the temperature 95C, while the oil temperature for frying is generally over 500C. They should provide the proof first that frying process did not harm DNA sequencing, or they should have used the samples from the market before they had been cooked. Except the sample, the methodology is fine, and all other parts were well written.
We agree with the reviewer that frying temperatures are high. Nevertheless, we believe temperature did not have an adverse effect on our particular sequencing results. In these high temperatures, DNA gets fragmented, making it harder to sequence whole genomes. But here, we only sequenced short fragments (Ben-Jerry region: ~400bp; LCO1490-HCO2198 region: ~700bp) of mitochondrial DNA, which is generally very abundant. The shorter the fragment the more likely it is to be picked up using PCR, as shown by Van der Colff & Podivinsky, Int J Food Sci Tech (2008), who tested the effect of different cooking techniques (raw, boiled, pressure-cooked, roasted, microwave and deep-fried) on the DNA quality in potatoes. They showed that DNA extractions were barely or not visible on electrophoresis gels for the treated samples, but depending on the length of the PCR fragment they were able to amplify fragments for all treatments, i.e. the shorter the fragment the easier it is. This is reflected in the fact that we were only able to amplify the longer 700bp fragments for only 4 of the samples, whereas we amplified the shorter 400bp fragment in all the 21 samples, as can be seen in the submitted alignment. While it would be interesting to compare fried and pre-fried samples, unfortunately, ‘hormigas culonas’ are only sold fried in the market. Nonetheless, in our dataset we also have ‘fresh’ samples of ants collected in Panama, as described in the Materials and Methods, which gave similar results as the fried Colombian samples, with the difference that it was easier to amplify the 700bp fragment. Given our results, we are confident that the sequences obtained are the correct ones and have not been affected by the frying process.
Here are numbers of suggestions, which might help to improve the manuscript.
Through manuscript, suggest to use the different English words for describing human society and ant ecology.
Ex. Authors use 'local population'(line16) to describe the human society, however it is better just to say 'local people'.
Done
In abstract
Please focus the topic to leaf-cutting ant as a food source. Covering topic about them as a pest, a forest depredator and a food source is not necessary in abstract.
We disagree with the reviewer. In the abstract we only briefly mention them being a pest as a way to show their contrasting effects on humans.
In introduction
Probably better without subtitles, or add subtitle to all paragraphs.
Agreed, subtitles removed.
Line 32-44, suggest to replace it to the discussion.
We disagree with the reviewer here. We think this paragraph is important to set the background for the manuscript and to justify the presented research.
Line 45-55, suggest to make the general explanation in 2-3 lines.
We have now reduced the text and incorporated it in the previous paragraph. Now L57-58 “Leaf-cutting ants have an obligate mutualistic relationship with the fungus Leucoagaricus gongylophorus (Möller) Singer (1986), originated approximately 18 MYA [8-11], and are now dominant herbivores in the Neotropics [12]. They are also notorious pests, with mature colonies consuming up to 500 kg (dry weight) plant material annually and have widespread economic impacts in local communities every year [13,14].” This also reduced the number of references suggested below by the reviewer.
In materials and methods
Line 108, 114, 115, please correct the reference style.
We followed the advice and regulations as stipulated in the International Code of Zoological Nomenclature (http://www.nhm.ac.uk/hosted-sites/iczn/code/) Chapter 11 Articles 51.2 and 51.3, which say “51.2. Form of citation of authorship. The name of an author follows the name of the taxon without any intervening mark of punctuation, except in changed combinations as provided in Article 51.3.” and “51.3. Use of parentheses around authors' names (and dates) in changed combinations. When a species-group name is combined with a generic name other than the original one, the name of the author of the species-group name, if cited, is to be enclosed in parentheses (the date, if cited, is to be enclosed within the same parentheses).” We, therefore, edited citations to have no parentheses for the scientific names on first appearance, unless the name comes from a recombination, for which parentheses are used. All citations are according to this Code as well as the http://antcat.org and http://www.speciesfungorum.org databases, which holds the most accurate citations for ants and fungi.
In discussion
Probably better without subtitles, or add subtitle for all paragraphs.
Agreed, subtitles removed.
In references
Suggest to reduce the references. There are too many references, which are only for general explanations of ecology of leaf-cutting ants.
As mentioned in a response above, by reducing and merging the first two paragraphs of the introduction we have now removed a number of these references.
Round 2
Reviewer 3 Report
The authors addressed most of my comments and did a good job revising the manuscript.